# Tuberculosis IRIS: Pathogenesis, Presentation, and Management across the Spectrum of Disease

**DOI:** 10.3390/life10110262

**Published:** 2020-10-29

**Authors:** Carson M. Quinn, Victoria Poplin, John Kasibante, Kyle Yuquimpo, Jane Gakuru, Fiona V. Cresswell, Nathan C. Bahr

**Affiliations:** 1School of Medicine, University of California, San Francisco, CA 94143, USA; 2Infectious Diseases Institute, College of Health Sciences, Makerere University, Kampala, Uganda; johnkasi93@gmail.com (J.K.); janegakuru18@gmail.com (J.G.); Fiona.Cresswell@lshtm.ac.uk (F.V.C.); 3Division of Infectious Diseases, Department of Medicine, University of Kansas, Kansas City, KS 66045, USA; vpoplin@kumc.edu (V.P.); nbahr@kumc.edu (N.C.B.); 4Department of Medicine, University of Kansas, Kansas City, KS 66045, USA; kyuquimpo@kumc.edu; 5Clinical Research Department, London School of Hygiene and Tropical Medicine, London WC1E 7HT, UK; 6Medical Research Council, Uganda Virus Research Unit, London School of Hygiene and Tropical Medicine Uganda Research Unit, Entebbe, Uganda

**Keywords:** tuberculosis, immune reconstitution inflammatory syndrome, tuberculous meningitis, acquired immunodeficiency syndrome

## Abstract

Antiretroviral therapy (ART), while essential in combatting tuberculosis (TB) and HIV coinfection, is often complicated by the TB-associated immune reconstitution inflammatory syndrome (TB-IRIS). Depending on the TB disease site and treatment status at ART initiation, this immune-mediated worsening of TB pathology can take the form of paradoxical TB-IRIS, unmasking TB-IRIS, or CNS TB-IRIS. Each form of TB-IRIS has unique implications for diagnosis and treatment. Recently published studies have emphasized the importance of neutrophils and T cell subtypes in TB-IRIS pathogenesis, alongside the recognized role of CD4 T cells and macrophages. Research has also refined our prognostic understanding, revealing how the disease can impact lung function. While corticosteroids remain the only trial-supported therapy for prevention and management of TB-IRIS, increasing interest has been given to biologic therapies directly targeting the immune pathology. TB-IRIS, especially its unmasking form, remains incompletely described and more data is needed to validate biomarkers for diagnosis. Management strategies remain suboptimal, especially in the highly morbid central nervous system (CNS) form of the disease, and further trials are necessary to refine treatment. In this review we will summarize the current understanding of the immunopathogenesis, the presentation of TB-IRIS and the evidence for management recommendations.

## 1. Introduction

*Mycobacterium tuberculosis* infection (TB) remains the world’s most deadly infectious disease with 10 million cases and nearly 1.5 million deaths in 2018 alone [1]. TB burden remains highest in parts of the world with high HIV prevalence—notable, given that TB and HIV co-infection negatively affects patient outcomes. TB infection leads to increased HIV replication and HIV contributes to TB progression due to HIV mediated immune suppression [2]. People living with HIV have a 19 times increased risk of developing TB, and make up 17% of all TB deaths worldwide [1]. TB accounts for one in three AIDS-related deaths, making it the leading contributor to mortality in people living with HIV [1].

The roll-out of anti-retroviral therapy (ART) in settings with high TB and HIV prevalence has been instrumental in combatting these twin epidemics, with ART availability reducing TB risk by 58–80% [3]. Since 2000, TB mortality in HIV-positive individuals decreased by 60% compared to 27% among HIV-negative individuals—almost certainly in part due to wider ART availability [1]. However, an important complication of TB-HIV coinfection in the post-ART era is TB-associated immune reconstitution inflammatory syndrome (TB-IRIS). First recognized in the years after the roll-out of ART, this syndrome is characterized by severe inflammatory features of TB occurring after rapid reconstitution of the immune system once ART has been initiated. This should be distinguished from TB paradoxical reaction (also sometimes referred to as IRIS though these are not the same phenomena), a long-recognized feature of the course of TB treatment where there is a brief paradoxical worsening of symptoms after initiating TB treatment, that is independent of rapid immune recovery caused by ART [4,5]. In 2006, the need to standardize terminology used for TB-IRIS [6,7,8] motivated the International Network for the Study of HIV associated IRIS (INSHI) to establish the currently-used consensus case definitions for the two different forms of TB-IRIS––paradoxical and unmasking (Figure 1) [4].

Diagnostic categories of TB-IRIS depend on the sequence of ATT (antitubercular therapy) and ART (antiretroviral therapy) initiation, as well as the site of TB at ART initiation. Paradoxical IRIS is diagnosed when pulmonary or extrapulmonary TB worsens after ART initiation in a patient who previously responded to ATT. ART-associated TB is any TB diagnosed after ART initiation, while unmasking TB-IRIS is a subcategory where the TB presentation is markedly inflammatory. We add a final major category of CNS-TB-IRIS based on its unique pathologic and therapeutic considerations.

Paradoxical IRIS is the onset of IRIS manifestations within three months of ART initiation in a patient previously diagnosed with TB and with an initial response to treatment. Manifestations include enlarging lymph nodes, cold abscesses, worsening radiological features, worsening serositis, or worsening of symptoms of at least two diseases locations (i.e., constitutional symptoms, respiratory symptoms, abdominal symptoms) [4].

ART-associated TB is active TB diagnosed after ART initiation in a patient not receiving TB treatment.

Unmasking TB-IRIS is ART-associated TB that occurs within three months of ART initiation and has a “heightened intensity of clinical manifestations, particularly if there is evidence of a marked inflammatory component”. Examples include, but are not limited to, lymphadenitis, abscesses, respiratory failure, or a systemic inflammatory syndrome. Importantly, in this case TB was not diagnosed or treated prior to initiation of ART [4].

The definition for paradoxical IRIS is well characterized and has been widely used, but the definition for unmasking IRIS remains provisional due to an unclear delineation of the inflammatory characteristics required, and therefore the term “ART-associated TB” has been used to include unmasking IRIS as well as non-IRIS TB diagnosed after ART initiation [4].

A clinically helpful framework for TB-IRIS is to classify each form by the patient population at risk for its development after ART-initiation—paradoxical TB-IRIS is seen in patients with active TB infection undergoing treatment, and unmasking TB-IRIS is seen in patients with occult or subclinical TB that generally has not been diagnosed. CNS-TB-IRIS is seen in patients with neuro-invasive TB and, while technically falling into one of the prior categories, is clinically and pathologically a unique entity with differing immune characteristics and treatment guidelines. In this review we will detail the current state of research on each form of TB-IRIS and describe the gaps in understanding that may serve as opportunities for future study. While TB-IRIS may occur in settings outside of HIV co-infection (for example, with a decrease in immune suppressive medications related to transplantation), the focus of this review relates to HIV [9].

## 2. Paradoxical TB-IRIS

Paradoxical TB-IRIS is the best characterized form of TB-associated IRIS, occurring in approximately 18% of TB-HIV coinfected patients, although individual studies have ranged from 4% to 54% [10]. This incidence might be somewhat lower in the pediatric population, where one study found approximately 7% [11].

### 2.1. Clinical Features and Diagnosis

There is no single diagnostic test for TB IRIS; diagnosis relies primarily on recognition of the sequential relationship between initiation of antitubercular therapy (ATT), ART initiation, and deterioration shown in Figure 1 [12]. Pulmonary and lymph node involvement are the most commonly reported signs albeit with wide reported ranges, present in 5.6%–88.5%, and 30%–100% of cases respectively [10,13,14,15,16]. Reported median durations of IRIS ranged from 19 to 87 days [10]. Elevated plasma C-reactive protein (CRP) supports a diagnosis of TB-IRIS [17,18]. Worsening of chest imaging is suggestive of IRIS, even when there is not clinical deterioration. Findings can include progression to miliary disease, worsening consolidations, enlarged thoracic lymph nodes, and pleural effusions. One study reported that in 45% of pulmonary TB patients, radiologic worsening occurred within one to five weeks of ART initiation and resolved within two weeks to three months [19]. Other imaging modalities can assist in diagnosing extrapulmonary TB-IRIS, including chest CT and mediastinoscopy for intrathoracic IRIS [20], brain CT and MRI scans for CNS-TB-IRIS [21,22], and ultrasound for intra-abdominal TB-IRIS [21].

Evidence of immune reconstitution, quantified by significant increase in CD4 count between ART-initiation and IRIS onset, further supports the diagnosis of IRIS. There is a 0.6–5.6-fold increase in CD4 T-cell count among patients with IRIS compared to the 0.2–1.0-fold increase among the non-IRIS group [23]. However in rare cases there is a decrease in absolute CD4 T-cell count at the time of IRIS diagnosis attributable to bone marrow suppression due to TB [20] and/or ATT [24].

Importantly, TB-IRIS often presents nonspecifically with systemic illness and lymphadenopathy and therefore INSHI definitions require investigations to rule-out potential IRIS mimics including drug reactions, treatment failure, superimposed infections, or malignancies [4]. Consideration of drug toxicities and interactions is important in HIV-TB coinfected patients, as the numerous adverse effects of ATT can be more frequent in those with HIV [25]. Suspected abdominal TB-IRIS should raise concern given the hepatic toxicities of ATT [26]. Failure of, and subsequent resistance to, both ART and ATT can lead to clinical deterioration, which can be indistinguishable from IRIS [27]. Poor adherence is especially common in those with TB and HIV [28], and a thorough exploration of barriers to treatment is important when nonadherence or IRIS is suspected. Multi-drug-resistant TB is an increasing problem in many countries with high HIV and TB burdens, although evidence suggests it is not more frequent in people living with HIV (PLHIV) [29]. Determination of drug resistance is best accomplished with mycobacterial culture, although molecular diagnostics such as Gene Xpert MTB/RIF Ultra can provide more timely diagnosis [30,31], at the cost of some missed cases of resistance [32]. Workup for alternate infectious etiologies should include blood cultures but can otherwise be guided by any localizing signs [4]; for example, worsening pulmonary symptoms should trigger consideration of *Pneumocystis jirovecci* pneumonia [33] or pulmonary Kaposi Sarcoma. In endemic regions, especially Latin America, HIV-associated histoplasmosis is often misdiagnosed as TB, and might have higher incidence [34]. Endemic non-opportunistic infections such as malaria can occur during TB treatment, and, based on the patient’s location, should be considered in cases of fever recurrence [35]. HIV patients with advanced disease are at a high risk for non-Hodgkin’s lymphoma, cervical or penile carcinomas, and Kaposi Sarcoma, which can present with lymphadenopathy mimicking TB-IRIS [36]. Biopsies of affected lymph nodes or other tissues with histopathologic examination can readily distinguish malignancies from the non-caseation granulomas of TB-IRIS [19].

### 2.2. Prognosis

While prognosis varies based on site of disease and treatment protocols, the course of TB-IRIS is most frequently self-limited [10,37]. The most recent metanalysis of TB-IRIS found a 7% all-cause mortality with 2% attributable to IRIS [10]. However, inconsistent criteria for defining TB-IRIS associated death and variant follow-up time have led to individual studies reporting anything from no deaths, to mortality as high as 38% [38,39,40,41,42,43,44,45,46]. Without standard glucocorticoid treatment, TB-IRIS was independently associated with 48-week mortality (aOR 2.72, 95%CI 1.14–6.54) [47]. There has been recent interest in the pulmonary morbidity associated with TB-IRIS. Approximately half of patients with HIV and pulmonary TB experienced clinically significant declines in the forced expiratory volume over one second FEV1 function after ART-initiation, and 18% experienced severe declines associated with long-term impairment [48]. In contrast, a sub-study of the predART trial (see “Management” section) found that the development of TB-IRIS was not associated with a significant reduction in pulmonary function with or without the use of prednisone [49].

### 2.3. Pathogenesis

TB-IRIS results from a disproportionate and dysregulated inflammatory response to TB produced as a consequence of rapid recovery of the immune system. The immunology of HIV-TB coinfection has been reviewed extensively [2,50,51,52]. TB bacilli proliferate in alveolar macrophages, where their detection by pattern recognition receptors triggers cytokine release and recruitment of other phagocytic cells to form granulomas [2]. In most immune-competent individuals, cell-mediated immunity controls proliferation and TB becomes latent. However, HIV leads to profound depletion and dysregulation of the CD4 T lymphocytes underlying cell-mediated immunity, and impaired signaling and phagocytosis in macrophages, so control of TB is not achieved [2]. TB-specific CD4 T cells are especially depleted due to increased expression of the cell surface chemokine receptor CCR5 (C-C chemokine receptor 5) [53], while antigen recognition [54] and cytokine production [55] are impaired in the remaining CD4 T cells. In addition, HIV is thought to inhibit TB-induced tumor necrosis factor alpha (TNF-α) release that triggers macrophage apoptosis and contributes to TB control [56]. Toll-like receptor (TLR) signaling involved in TB recognition is decreased in HIV infection as well [57]. With the aforementioned impairments of macrophages and CD4 T cells in advanced HIV infection, TB-exposed individuals are more likely to have significant TB replication and broadly disseminated disease [58]. In paradoxical TB-IRIS, ART initiation leads to a restoration of many aspects of the immune response to TB, while others remain impaired. For example, CD4 T cell count may increase, but ratios of memory to naïve subsets remain unbalanced [59]. This leads to a pathologic disproportionate inflammatory response against TB which are already dead or dying as a result of ATT.

There has been extensive work characterizing the inflammatory response in TB-IRIS. While animal models were essential in understanding TB pathogenesis [60], models for TB-IRIS have been limited by the inability of HIV to infect non-primates. One group has shown promise with a *Mycobacterium avium* IRIS mouse model [61,62] and recent advances in humanized mouse models of HIV-TB coinfection could be adapted for studying IRIS [63]. The current literature on TB-IRIS has therefore relied on analyses of human samples, including plasma cytokine measurements, transcriptome analysis, and in vitro stimulation of peripheral blood mononuclear cells (PBMCs), and has broadly supported a central role of a synergistic interaction of the innate and adaptive immune systems by way of the self-reinforcing responses of dysregulated CD4 T cells and macrophages in the unchecked inflammation of TB-IRIS (Figure 2).

TB-IRIS is characterized by exuberant release of interferon- γ (IFN-γ) and related cytokines by mycobacteria-specific Th1 type CD4 T cells which expand in number and responsiveness with ART [16,64,65,66]. ART also causes an increase in cytokines related to Th1 type CD4 T cells such as Interleukin (IL)-2, IL-12, IFN-γ, IFN-γ inducible protein 10 (IP10), IL-18, Tumor necrosis factor alpha (TNF-α), and soluble IL-2 receptor (sCD25) [65,67,68,69,70]. Another study further demonstrated pathogen-specific dysregulation by showing that an increased response to TB antigens was accompanied by decreased response to non-TB antigens and that overall T cell quality was poor [66]. However, other studies have questioned whether this increased Th1 response explains IRIS, as non-IRIS patients also had Th1 T cell expansions (albeit smaller than IRIS patients) [16] and IFN-γ responses have been found to differ based on ART status but not IRIS status in another study [71]. A distorted subset balance also contributes to IRIS, with an increase in effector-memory CD4 T cells [72,73], and a decrease in central-memory CD4 T cells [73] in TB-IRIS patients.

There is increasing evidence that the innate immune component of IRIS centers on aberrant activation and signaling of macrophages and monocytes. Lai et al. showed, using whole blood transcriptomics, that macrophage signaling was the initial upregulated process after ART-initiation [68]. Various markers of monocyte activation (soluble CD14, soluble CD163, soluble tissue factor) are increased at IRIS onset, and unbalanced monocyte populations are characteristic of IRIS and closely paralleled inflammatory markers [74]. Histopathology implicates macrophages as the primary inflammatory cell in some TB-IRIS cases [75]. Numerous studies report differences in expression of the pattern-recognition receptors that begin the inflammatory cascade [76], including toll-like receptor (TLR) signaling [68,77], and inflammasome activation [68,78]. TLR signaling is increased in TB-IRIS [68], with increased TLR2 expression on monocytes as well as myeloid dendritic cells, paralleling production of downstream cytokines (TNF-α, IL-12p40) [77]. Inhibition of the downstream adaptor of TLR2 (myeloid differentiation primary response 88 or MyD 88) decreased the secretion of those downstream cytokines in IRIS patients [68]. Persons with IRIS also showed increased transcription and secretion of multiple elements of inflammasome activation (Triggering receptor expressed on myeloid cells-1 (TREM-1), IL-1, caspase-1 and -5), and found that IL-1 secretion was dependent on inflammasome activation [68]. Studies have also found increased IL-18, an inflammasome cytokine, in TB-IRIS [79,80]. Increased expression of complement-related genes, including the C5 gene, in macrophages and monocytes suggests a role of the complement system in the innate immune response of IRIS [68,76,81].

Given the downstream effects of increased Th1 cell and macrophage activation on the production of IFN-γ and TNF-α respectively, it is likely that these systems together produce the exuberant cytokine response characteristic of TB-IRIS. IL-6, the cytokine responsible for initiating systemic inflammation, has been one of the most consistently shown biomarkers of TB-IRIS [14,64,68,69,70,82], alongside its downstream acute phase reactant, CRP [14,18,83]. IRIS is associated with a range of other cytokine elevations, further contributing to this storm [64,68,69,70,80,84]. Corticosteroid treatment has been shown to lower IL-6 and TNF-α, providing a mechanism for their efficacy in treating IRIS [69].

Neutrophil chemotaxis is driven by this inflammatory milieu, especially the IL-1 product of inflammasome activation, causing infiltration at the site of TB-IRIS disease [85]. Systemically, TB-IRIS is associated with upregulated transcripts of neutrophil activation genes (S100 calcium binding protein A9 (S100 A9), the NLR family pyrin domain containing 12 (NLRP12), cyclooxygenase 1 (COX-1) and IL-10) and increased blood levels of neutrophils, elastase, and human neutrophil peptides [85]. Other studies have also found overexpression of S100 A8 and A9 [86], including in TB meningitis IRIS [87], where neutrophils are especially important [78,87]. Matrix metalloproteinases (MMP), endopeptidases involved in tissue repair and remodeling, are also increased in TB-IRIS, in particular MMPs 1, 3, 7, 8, 9, and 10 [87,88,89,90]. Two of these studies found evidence that MMP secretion was driven by neutrophils [87,89], with elevated extrapulmonary extracellular matrix turnover (the effect of MMP activation) also being associated with IRIS, providing a possible explanation for their role in the inflammatory response. MMP-7 decreased following corticosteroids [88].

Four distinct lineages of lymphocytes with cytotoxic potential have been investigated in TB-IRIS. There have been mixed findings on CD8 T cells, with evidence of decreased cell-signaling in IRIS patients [66], decreased activation pre-ART but not at IRIS onset [91], increased expression of cell-surface receptors killer cell ligand like receptor g1 (KLRG1) and programmed cell death protein 1 (PD-1) [92]; a study also found elevated CD8 T cells as a risk factor for TB-IRIS, with expansion of two subpopulations at IRIS onset [93]. γδ T cells are another distinct T cell population found to be more activated in TB-IRIS [72,94], although their exact role is yet to be fully elucidated. In TB-IRIS there is upregulation of perforin and granzyme B, cytotoxic granule components of invariant Natural Killer T cells (iNKT), a unique T cell population thought to bridge adaptive and innate immune responses [95]. A recent study by Walker et al. using flow cytometry found an increased frequency of iNKT cells in TB-IRIS with evidence of recent degranulation, but a paucity of the immunoregulatory subset of iNKT cells [96]. Other studies have found increased natural killer cell activation [83] and increased degranulation capacity [97] in TB-IRIS patients.

Humoral immunity does not appear to significantly contribute to the inflammatory response of TB-IRIS [98,99], Similarly, Th2 response are notably generally absent in the inflammatory response of TB-IRIS [65].

Together, these studies support a model of TB-IRIS whereby early post-ART changes in innate immune signaling in macrophages, and dysregulated CD4 T cell recovery, lead to hypercytokinemia with systemic and local inflammation (Figure 2). Neutrophils and cytotoxic lymphocytes are important mediators of the inflammation and tissue damage. These and continued explorations of the immune mediators of TB-IRIS provide numerous avenues for selecting future therapeutics and predictive algorithms which can have real impacts on clinical outcomes for these patients.

### 2.4. Management

Evidence for TB-IRIS management is limited (Table 1), but corticosteroids are first-line therapy (prednisone 1.5 mg/kg/day tapered over four weeks) [10,100,101,102,103]. The duration of prednisone may need to be extended due to recurrence of symptoms with weaning or discontinuing steroids, but continuation beyond four–six months is discouraged [101,102,103,104]. Evidence for corticosteroids is provided by a South African trial that randomized patients with moderate to severe, non-life-threatening TB-IRIS to receive either prednisone (dosed as above) or placebo [100]. This trial found that subjects who received prednisone had decreased hospital length-of-stay, and more rapid improvement in symptoms, inflammatory markers, and quality of life indicators, without an increase in severe infections [100].

Nonsteroidal anti-inflammatory drugs (NSAIDS) have been used for symptomatic management of mild or localized TB-IRIS, but have not been evaluated in RCTs [10,102,104,105,106,107]. Biologic and traditional immunomodulators including thalidomide, TNF-α inhibitors, IL-6 blockers, montelukast, and pentoxifylline have been used for treatment of refractory TB-IRIS in case reports, but as yet there is inadequate evidence to recommend any of these for routine use (Table 1) [101,102,104,105,106,108,109,110].

Procedures including lymph node aspiration, pericardiocentesis, paracentesis, and thoracentesis are sometimes required to alleviate symptoms or manage complications [101,104,106,107,108]. TB-IRIS is not an indication to extend ATT, except in patients with abscesses or tuberculomas that have persisted despite six months of TB treatment in which ongoing active infection is suspected [101]. Importantly ART should be continued, except when there is life-threatening neurologic disease (see section on CNS-TB-IRIS) [10,101].

### 2.5. Risk Factors and Biomarkers

Quantification of risk for IRIS development allows the targeting of prevention to those at greatest risk (Table 2). Given our understanding of IRIS as a robust immune recovery targeting TB antigens, it is not surprising that most published studies point to those with disseminated TB being at highest risk of developing paradoxical TB-IRIS [13,23,119] along with those with advanced HIV [10]. Low CD4 count, high viral load, and disseminated TB are frequently coincident and associations do not consistently hold up in multivariate analyses—thus, it is not clear which factor might be most important [10]. However, the importance of CD4 count is supported by numerous studies [23,38,120,121]. Multiple trials also support viral load as a risk factor for IRIS development [13,38,39,122]. In resource limited settings, urine antigen tests for the TB cell wall glycoprotein TB-lipoarabinomannan (LAM) may assist in risk-stratification, as detection of LAM antigen is associated with both low CD4 count and disseminated TB [123]. Studies in both South Africa and Uganda have shown an association between pre-ART LAM positivity and subsequent TB-IRIS, although there is no evidence that this performs better than CD4 count [39,89].

While there is great interest in an immune signature of TB-IRIS, no biomarker has yet been validated as a clinically useful predictor. Most promising have been studies which combined multiple inflammatory markers to predict risk, with many including CRP, IL-18, and IL-6 [14,17,80,81,90,124] (Table 2). A recent metabolomic profiling study generated an eight biomarker predictor including components of arachidonic acid, linoleic acid and glycerophospholipid metabolism which could distinguish TB-IRIS from non-IRIS with sensitivity of 92% and specificity of 100%, although this lacked a separate validation cohort [125].

Two small studies have also suggested genetic risk in IRIS development, with polymorphisms in the IL-6, IL-18, TNF-α, vitamin D receptor, and natural resistance-associated macrophage protein 1 genes associated with differential risk for TB and *Mycobacterium avium* complex IRIS [126,127]. A recent study of 88 TB/HIV coinfected patients in Brazil showed increased risk of TB-IRIS in patients with specific HLA haplotypes and polymorphisms in a gene for NK cell receptors [128].

### 2.6. Prevention

In patients with TB and CD4 counts of less than 100 cells/µL, prednisone for four weeks is recommended based on one large clinical trial for prevention of TB-IRIS [101,130]. The Pred-ART randomized subjects to placebo or prednisone (40 mg for two weeks followed by 20 mg for two weeks) and found a 30% reduced incidence of TB-IRIS (RR 0.7) in persons with CD4 counts less than 100 cells/µL with prednisone [131]. Maraviroc, an antiretroviral that inhibits CCR5, has been proposed for prevention of IRIS but was not effective in a placebo-controlled trial [131].

In patients with TB who are subsequently or concurrently diagnosed with HIV, the timing of starting ART is a key factor in the risk of development and severity of IRIS risk given IRIS is associated with the antigen burden at the time of immune recovery and CD4 cell counts [105,132]. Metanalyses show a greater than two-fold increased risk of IRIS and a seven-fold (RR = 6.94; 95% CI = 1.26–38.22) increase of TB-IRIS associated death with early (1–4 weeks) ART initiation [133,134]. However, the decreased risk of IRIS with delayed ART must be balanced against the increased risk for opportunistic infections and mortality in patients with severely reduced CD4 cell counts. In the aforementioned metanalyses, early ART reduced mortality (RR 0.81, 95% CI 0.66 to 0.99) despite the increase in IRIS [134]. In a subgroup analysis, the effect size was greater in those with CD4 < 50 cells/µL (RR, 0.71; CI 0.54 to 0.93), but no difference was found in those with CD4>50cells/µL (RR, 0.71; CI 0.54 to 0.93) [134]. Based on this it has been recommended to initiate ART within 2–4 weeks in individuals with CD4 count <50 cells/µL and to delay ART to 8–12 weeks in patients with CD4 counts >50 cells/µL [135]. Guidelines differ for TB meningitis, as will be expanded upon below [103,136].

## 3. CNS-TB-IRIS

Given the physical and immunologic peculiarities of the central nervous system, paradoxical IRIS in the setting of CNS TB requires specific considerations in diagnosis and management. CNS TB-IRIS is common, occurring in nearly half of TB meningitis (TBM) cases in some cohorts [136], and representing the most common etiology of neurological deterioration after ART initiation [137]. However, despite making up about 10% of all TB-IRIS [138], neurological manifestations contribute to the vast majority of TB-IRIS mortality [10,46]. The prognosis of CNS-TB-IRIS is poor with studies showing mortality in 13–30% of cases and disability in an additional 30% [136,138]. Of note, despite paradoxical reactions also being common in HIV-negative TBM, these non-IRIS paradoxical reactions do not adversely impact mortality [139], and therefore parallels between these entities should be made cautiously.

### 3.1. Clinical Features and Diagnosis of CNS-TB-IRIS

As with all paradoxical TB-IRIS, CNS-TB-IRIS is diagnosed when symptoms of CNS-TB recur after ART-initiation after initially improving on ATT (Figure 1) [4]. Symptoms reappear a median of 14 days after ART initiation [136,138]. CNS TB-IRIS can include all forms of CNS TB—most prominently meningitis, but also intracranial tuberculoma [22], brain abscess [140,141], spinal epidural abscess, and radiculomyelitis [136,138]. The most common symptoms are headache, neck stiffness, confusion and new-onset seizures [136,138]; paraparesis, disconjugate eye movements, aphasia, and decreased vision are also reported [136,139]. Systemic signs of IRIS and TB, such as lymphadenopathy, are also instructive [139]. Inflammation in the CSF typically coincides with symptoms, improving initially during ATT, but then returning to the range at initial presentation; specific thresholds of leukocytes, protein, or glucose have not been established [136,138]. Neutrophils are often predominant at IRIS onset [136,139].

Head and spine imaging are also important for diagnosis as radiologic worsening is itself sufficient for diagnosis of IRIS in some situations [4], and significant hydrocephalus, which is often present in CNS-TB-IRIS, should trigger intervention [139]. Tuberculomas are the most common imaging finding in CNS-TB-IRIS, occurring in 52% of cases in one series, often alongside meningitis [138]. They are generally supratentorial, moderately sized (median 11 mm), and multiple in half of cases, usually with associated edema, but rarely with mass effect [22]. The presence of multiple lesions and perilesional edema is more common in patients with IRIS than at initial TBM presentation [22]. Presentations localizing to the spinal cord should trigger spine imaging to diagnose radiculomyelitis or epidural abscess [136,138].

While neurologic worsening after ART-initiation in a patient being treated for TBM is strongly indicative of CNS-TB-IRIS, several other conditions should be considered. When CSF culture or molecular diagnostics are positive, it is important to rule out drug-resistant TB, which can occur in more than 10% of TBM cases and is associated with poor outcome [142]. Cryptococcal meningitis, toxoplasmosis, CMV encephalitis, HIV encephalopathy or progressive multi-focal leukoencephalopathy may be present as co-infections with TBM and may be unmasked due to ART [137,138].

### 3.2. Pathogenesis of CNS-TB-IRIS

Although the immunopathogensis of TB-IRIS in the CNS is likely to mirror that of IRIS outside of the CNS, there are important differences. Many TBM-IRIS associated cytokines, chemokines, and mediators involved in neutrophil response or inflammasome activation are found in the CSF of TBM-IRIS patients in high concentrations, but blood concentrations do not differ significantly between TBM patients with and without IRIS, suggesting a highly compartmentalized inflammatory response [87]. Despite this divergence in apparent inflammatory response, transcriptional profiling has shown expression of similar genes in the blood and CSF [78], thus gene expression does not explain the difference between CNS and peripheral immune responses. Transcripts associated with neutrophil response and both canonical and non-canonical inflammasomes are found at higher concentrations in the blood of patients with TBM-IRIS than in TBM patients without IRIS. S100 A8/A9 (unlike other inflammatory mediators) is elevated in TBM-IRIS even after controlling for baseline culture positivity, which is an independent risk factor for the development of TBM-IRIS [78,87,136].

This understanding of the immunopathology of TBM-IRIS has encouraged efforts to better predict which TBM patients will go on to develop IRIS (Table 2). As in extra-meningeal TB-IRIS, high viral load and low CD4 T cell count at initial presentation remain risk factors. Further, at TBM diagnosis, increased CSF neutrophil counts (AUC 0.72, 95% CI 0.54–0.90) and CSF culture-positivity (RR 9.3, 95%CI 1.4–62.2) predict future development of paradoxical TBM-IRIS [136]. One model also found high CSF TNF-α and low IFN-γ at TBM diagnosis to be predictive (AUC 0.91, 95%CI 0.53–0.99) of CNS-TB-IRIS [136]. Marais et al. have also used transcriptional analysis to predict IRIS development in TBM, with specific elevation in transcripts associated with neutrophil activation, especially matrix metalloproteinases [78]. These predictors have yet to be validated but show promise in future efforts to tailor both ART timing and immunomodulatory therapies to those patients at most risk of CNS-TB-IRIS.

### 3.3. Prevention of CNS-TB-IRIS

Protocols for prevention of IRIS are where TBM-IRIS differs most from other forms of TB-IRIS. Unlike in extra-meningeal TB, the high morbidity and mortality of CNS-TB-IRIS makes the risk of IRIS related to early ART initiation higher than the risk of developing additional infections related to delaying ART. The American Thoracic Society and Centers for Disease Control recommend delaying ART initiation until 8 weeks after ATT initiation [143], and other expert opinions recommend at least waiting several weeks from ATT initiation, irrespective of CD4 count [101]. Randomized, controlled trial evidence for this recommendation is provided by a Vietnamese study that showed initiating ART within one week was associated with a significant increase in grade 4 adverse events [135]. U.S. CDC guidelines also recommend 8 weeks of adjunctive corticosteroids with ATT based on a Cochrane metanalysis demonstrating reduced mortality when adding corticosteroids (RR 0.75, 95% CI 0.65 to 0.87) [144]. The rationale for adjunctive steroids is the dampening of pathologic immune responses, of which IRIS is one. This review, however, included only one study with HIV-positive patients (*n* = 98), and therefore was not able to definitively conclude whether mortality was improved in this subpopulation (RR 0.90, 95%CI 0.67 to 1.20), or to assess incidence of IRIS [144]. There is concern that corticosteroids may in fact lead to poor outcomes in TBM in persons with HIV and an ongoing clinical trial will provide a more definitive answer this question (NCT03092817). Several trials are currently underway to assess intensified antitubercular therapy. Given the association between inadequate mycobacterial killing and TBM-IRIS, these regimens may have an impact on IRIS prevention although whether they will improve TBM outcomes (their goal), let alone prevent IRIS, is unclear [145]. One trial of intensified ATT in TBM published thus far did not improve outcomes overall or in HIV infected subjects specifically—IRIS was not specifically reported on [146].

### 3.4. Management of CNS-TB-IRIS

While many immunomodulating therapies have been tried with some success in treating CNS-TB-IRIS once it has developed, high-quality evidence is lacking (Table 1). In fact, many therapeutic trials for TB-IRIS specifically exclude patients with CNS-TB IRIS [100,130]. In one observational study, 18 of 21 patients receiving corticosteroids for CNS-TB-IRIS showed initial improvement (median 10 days after administration) [138]. Prednisone (1.5mg/kg/day) did not prevent TBM-IRIS in a non-controlled cohort of 34 TBM patients, and dose-increase or restarting of prednisone at IRIS diagnosis decreased the concentration of only two inflammatory mediators (G-CSF, IL-37), while other markers of inflammation remained elevated from ART initiation [87]. This previously-noted finding of limited CNS impact of corticosteroids [147] is in contrast to non-CNS TB-IRIS, where the peripheral immune response is modulated by corticosteroid administration [70,148]. This differential corticosteroid response suggests that more potent or focused immunomodulating therapy might be necessary for TBM-IRIS, whether due to differences in the immune response to CNS TB, or to challenges in reaching adequate CSF concentrations. Several case reports have reported success treating severe cases of CNS-TB-IRIS with TNF-α inhibitors such as infliximab and adalimumab, generally after failure of corticosteroids [109,112,113,114]. Thalidomide has been used with some effect for treatment of otherwise intractable intracranial tuberculomas [116,117]. However, given thalidomide’s teratogenicity, and the early stoppage of a randomized trial in pediatric TBM due to adverse events in the Thalidomide group [149], it should be reserved for severe cases that are refractory to corticosteroids [106,115].

## 4. Unmasking TB-IRIS

Unmasking TB-IRIS is an increasingly recognized complication of initiating ART in a patient with subclinical or occult TB infection (Figure 1). Many epidemiologic studies demonstrate that TB is most commonly diagnosed in the first three months after ART-initiation [150,151,152,153,154]. Unmasking TB-IRIS is seen in 1–4% of patients initiating ART [41,151,155,156], with 23–37% of those with ART-associated TB being classified as unmasking IRIS [151,157]. However, a poor understanding of what constitutes an inflammatory phenotype in the absence of a proven biomarker for IRIS makes definitive differentiation of unmasking IRIS from non-IRIS TB presenting after ART initiation difficult.

### 4.1. Pathogenesis and Immunology

The pathogenesis of unmasking TB-IRIS is not fully elucidated, but its final immunologic response is thought to involve similar mechanisms to the uncontrolled immune response that occurs after rapid immune restoration in paradoxical IRIS [2]. Subclinical and occult TB is common in HIV infected patients not on ART [158,159], suggesting that the dampened immune response leads to minimal symptoms of TB, which can be difficult to distinguish from the symptoms of advanced HIV. If ART is initiated while TB remains undetected, unmasking IRIS can occur as immune-reconstituted patients mount an exaggerated and dysregulated response to TB which, without prior ATT, is more likely to be actively replicating, and therefore more widespread, than in paradoxical IRIS [154]. Tuberculin skin test site biopsies from unmasking TB-IRIS patients show that these dysregulated immune responses to TB persist even once CD4 counts have recovered and are characterized by attenuated IL-10 concentrations and other features associated with Th2 and neutrophilic responses [160]. Other immunologic studies of unmasking TB-IRIS have demonstrated the importance of innate immune effectors such as NK cells, and the balance between central memory and effector memory T cells prior to IRIS onset [83,161]. By comparison, non-IRIS ART-associated TB would result from either a development of primary TB after ART-initiation (without memory T cells to MTB, there could not be rapid immune response) [154], or a more measured immune response to reactivated TB, perhaps due to lower bacillary burden or less profound immunosuppression at ART initiation. Further research related to the immunologic differences between unmasking IRIS and ART-associated TB is needed to allow for clinical correlation and more precise diagnosis.

### 4.2. Clinical Presentation and Management

The presentation of unmasking TB-IRIS is similar to the paradoxical form, though disseminated TB is more common [151,155,157]. Published cohorts describe presentations characterized by weight loss, lymphadenopathy, meningism, severe pneumonia, and tuberculous abscesses [151,155,157]. Onset has been reported from as soon as 4 days after ART initiation [157] with the largest South African cohort reporting a median of 12 days (IQR 7–49) [41]. Diagnosis of unmasking TB-IRIS can be more challenging than paradoxical TB-IRIS as the symptom trajectory is often not as clear. TB meningitis, a common presentation of ART-associated TB [155,157], is typically labeled as IRIS since it is a severe extrapulmonary manifestation, but the degree of inflammation can be variable in TBM [162] and there are no recognized thresholds of CNS inflammation for diagnosis as unmasking IRIS. Further complicating diagnosis is the fact that multiple opportunistic infections can also be unmasked in a single patient; for example, several cases of unmasking TB-meningitis IRIS with cryptococcal meningitis coinfection have been reported in a Ugandan cohort [163].

Management strategies are not well defined or evidence-based for unmasking IRIS. While earlier studies describe withdrawal of ART in severe cases [157], more recent guidance has recommended continuation [164]. Use of corticosteroids in unmasking IRIS has not been well studied but corticosteroids are used to control severe inflammatory manifestations [157]. Prognosis of unmasking TB-IRIS is difficult to estimate, as most cohorts have relatively few cases, with varying degrees of severity. One large South African cohort had a 16% case fatality rate and a study of ART-associated TB had a 20% mortality rate [41,150], while other studies found lower mortality attributable to unmasking IRIS [151,157].

### 4.3. Risk Factors and Prevention

Careful history taking, physical examination, and review of chest radiography at the time of ART initiation is important to prevent unmasking IRIS [154,165]. Risk factors for unmasking TB-IRIS include low baseline CD4 count and high viral load, as in paradoxical IRIS [157,166]. Signs of subclinical TB such as anemia, significant weight loss, and high CRP are predictive of developing unmasking IRIS and lymphadenopathy on chest x-ray imparts a nine-fold increased risk of developing unmasking IRIS [41,151]. Those patients with suspected TB in whom ATT is not started are at particular risk [166]. Furthermore, tuberculin skin tests and interferon gamma release assays are often negative in cases of severe immunosuppression and so are unreliable in diagnosis of latent TB in such settings with implications for initiating treatment prior to ART initiation [167,168].

Point-of-care screening assays have been developed to identify those patients with extrapulmonary TB that might be otherwise missed. The urine lateral flow assay for LAM has become an attractive option to diagnose disseminated TB in people with HIV due to their ease of incorporation into HIV clinics for rapid screening [123]. While sensitivity is poor in immune competent patients, it is improved in those severely immunosuppressed patients most at risk for unmasking IRIS—pooled sensitivity was 17% (IQR 10–27) among those with CD4 > 100 cells/µL versus 54% (IQR 38–69) among those with CD4 < 100 cells/µL [169]. In 2019, the WHO updated its guidelines to recommend outpatient LAM testing for anyone with a CD4 count of less than 100 cells/µL irrespective of symptoms [170]. While LAM testing has been shown to improve diagnostic yield for TB [171], it is unclear if it can detect cases not already suspected for TB, and small studies have shown it unable to predict either unmasking or paradoxical IRIS [121,172]. A novel LAM assay has increased sensitivity and thus might allow prevention of unmasking IRIS [123]. Tests of both urine and sputum with GeneXpert MTB/RIF Ultra can also improve detection of TB in PLHIV, and return results quickly enough to allow for providing ATT at the time of ART-initiation [173,174,175], potentially allowing prevention of unmasking IRIS [158].

Prophylactic TB treatment for patients initiating ART has become more widespread given WHO recommendations for combined prophylaxis with trimethoprim-sulfamethoxazole and isoniazid/pyridoxine [176]. While this is intended to treat latent TB infection and thereby prevent reactivation in those with advanced HIV [177], recent trial evidence also suggests potential to prevent unmasking TB-IRIS. A multi-site trial in sub-Saharan Africa showed decreased IRIS events and deaths in those taking enhanced prophylaxis (this included trimethoprim–sulfamethoxazole plus at least 12 weeks of isoniazid–pyridoxine, 12 weeks of fluconazole, five days of azithromycin, and a single dose of albendazole) as compared with standard prophylaxis (trimethoprim–sulfamethoxazole alone). Clearly this effect was not from prevention of TB alone; prevention of *Cryptococcus* meningitis IRIS in particular is a notable factor [178]. Several studies have assessed the value of empirical TB treatment compared to extensive screening or isoniazid preventative therapy in patients with advanced HIV and found no difference in major outcomes, suggesting that with truly rigorous screening it is possible to detect all but latent TB, although unmasking IRIS was not specifically assessed [179,180,181]. The role of TB preventative therapy in preventing unmasking TB-IRIS, and its implications for drug-resistant TB, require further exploration.

## 5. Conclusions and Future Directions

TB-IRIS in all forms causes significant morbidity and mortality in persons with HIV/TB coinfection and requires special focus given its partially iatrogenic nature. While significant strides have been made recently, both in the realms of immunology to better understand pathogenesis, and in clinical trials to better optimize prevention and treatment strategies, we are still lacking the knowledge we need to avoid poor outcomes of TB-IRIS—particularly in IRIS related to TBM. As the arsenal of targeted immunosuppressive medications expands, there is newfound opportunity to target our treatments of TB-IRIS to its known immunologic mediators, such as IL-6 or the inflammasome. This will require continued research to better characterize the immune response, as well as rigorous trials to demonstrate improved efficacy and decreased side effects compared to corticosteroids.

Unmasking TB-IRIS remains incompletely described and its prevention will require improved TB detection at ART initiation. As availability of ART and implementation of HIV test and treat continues to increase, ability to rapidly rule out TB coinfection is important. Continued research to identify IRIS biomarkers is of particular importance in unmasking IRIS given its diagnostic challenges.

TB-IRIS in the CNS must be a priority for further research given its high mortality. CSF laboratory tests have been shown to predict TBM-IRIS development, which opens an avenue for targeting advanced immunosuppressants or intensified TB treatment to those at highest risk, but such approaches require clinical trials. Given the localized disease characteristic of CNS-TB-IRIS, it is possible that unique immune-targeted therapies might be beneficial for CNS-TB-IRIS compared to disease outside of the nervous system.

## Figures and Tables

**Figure 1 life-10-00262-f001:**
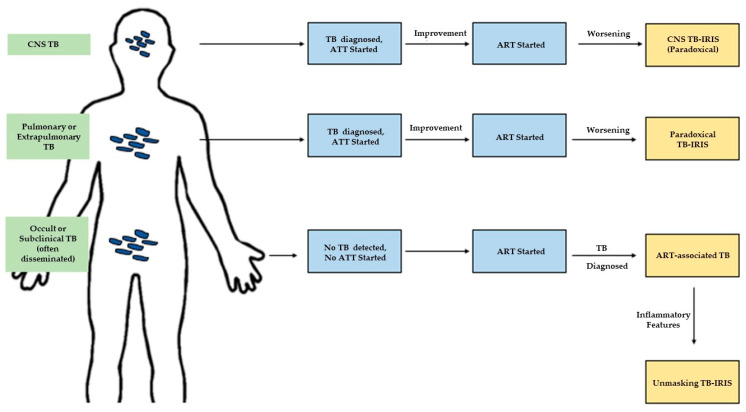
Diagnostic categories of TB-associated immune reconstitution inflammatory syndrome (TB-IRIS).

**Figure 2 life-10-00262-f002:**
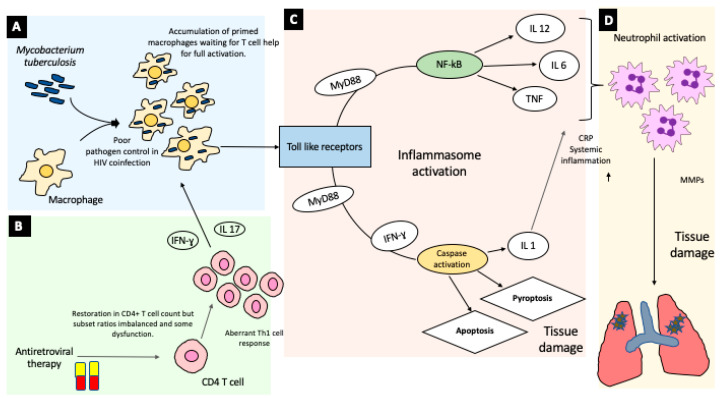
Model of TB-IRIS Pathogenesis. (**A**). HIV-related immune suppression leads to proliferation of TB in macrophages. (**B**). Antiretroviral therapy leads to increasing CD4 T cell counts, while the Th1 cell response remains dysregulated. Interferon signaling leads to an exuberant immune response against TB-infected cells. (**C**). This primed immune response to TB involves activation of Toll-like receptors, which leads to a two-pronged inflammatory cascade involving the inflammasome., with subsequent caspase activation and tissue damage, as well as activation of cytokines including TNF, IL6, and IL12. (**D**). Neutrophils are activated by this inflammatory cascade, especially IL-1, and they migrate to affected organs, leading to tissue damage mediated by matrix metalloproteinases (MMPs).

**Table 1 life-10-00262-t001:** Summary of evidence for TB-IRIS treatment regimens.

Treatment	Evidence
NSAIDs	No formal trials, based on symptomatic improvement in mild cases
Corticosteroids	Randomized placebo-controlled trial that found more rapid improvement in symptoms, radiography, inflammatory markers, performance and quality of life with four week course of prednisone [100]
TNF-α Inhibitors	Two case reports found clinical improvement with infliximab in patients with TB-IRIS refractory to corticosteroids [109,111] Two case reports of adalimumab for refractory CNS TB-IRIS [112,113]
Thalidomide	Case report with three cases of steroid dependent IRIS (2 cryptococcal meningitis and one disseminated TB) that had clinical improvement and allowed steroid tapering [114] Case reports showing clinical and radiographic improvement of intracranial tuberculomas [115,116]
Montelukast	Case report of three cases of severe IRIS (one due to secondary syphilis and two related to tuberculosis) that saw clinical improvement with montelukast [110]
Pentoxifylline	Case report of one case of TB-IRIS that found clinical improvement [117]
VEGF Inhibitors	Case report of Bevacizumab used for CNS-TB-IRIS (retinal) [118]

**Table 2 life-10-00262-t002:** Major findings regarding TB-IRIS risk factors and predictive biomarkers.

Risk Factor (at Initial TB Diagnosis)		Context	Risk Statistics
Standard Baseline Labs	Low CD4 T cell count	TB IRIS risk: CD4 < 50:23.1% (95% CI: 15.1–33.8) CD4 50–200:12.3% (95% CI: 8.7–16.8) CD4 > 200: 5.6% (95% CI: 3.2–9.4) [38] 8 studies with association reviewed by Namale, et al. [10]	RR 4.1 [38]
	High HIV Viral Load	Viral load >100,000 copies/mL associated with TB-IRIS 7 studies reviewed by Namale et al found association [10]	RR = 2.7 [38]
	Positive urine LAM	Urine LAM positivity predicted TB-IRIS, but not after controlling for CD4	OR, 10.9 (95% CI, 1.02–115.88) [89] OR: 4.6 (95% CI: 1.5–13.8) [121]
Cytokines and Acute-Phase Reactants	Elevated C-reactive protein (CRP)	Cutoff used median value	AUC = 0.73, sensitivity = 76% [14]
	Elevated IL-6	Cutoff used median value	AUC = 0.7, sensitivity = 80% [14]
	Elevated IL-6 and CRP	Cutoff used median value	Sensitivity = 92% [14]
	Elevated IL-18	Reduced, but still significant predictor in validation cohort (AUC = 0.742)	AUC = 0.99 [79]
	CXCL-10	Did not remain a significant predictor in the validation cohort	AUC = 0.884 [79]
	Elevated CRP, IFN-λ, sCD14, and low Hgb	Inflammatory score predicted IRIS vs non-IRIS	AUC = 0.82 [17]
	IFN-λ response, CCL-2, CXCL-10, and IL-18	IFN-λ response alone was a poor predictor (AUC = 0.61)	AUC = 0.9 [124]
Metabolite Profile	8 Metabolic biomarkers	Pathway analysis of biomarkers showed primary contribution of arachidonic acid, linoleic acid and glycerophospholipid metabolism	Sensitivity = 0.9, specificity = 1.0 [125]
Genetic Poly-Morphism	IL-6, TNF-α genes	Mycobacterial IRIS associated with IL6-174 * C carriage and TNFA-308 * 1 non-carriage [126]	Combined RR = 3, P = 0.014
	Cytokine genes	In Cambodian patients, TNFA-1031*T and SLC11A1D543N *G were associated with IRISIn Indian patients, IL18-607 * G and VDR FokI(F/f)*T were associated with IRIS [127]	P = 0.05, OR = 3.6; P = 0.04, OR = 0.21; resp] P = 0.02, OR = 3.8; P = 0.05, OR = 3.3, resp
	HLA alleles, NK cell receptor genes (KIR2D)	increased risk in carriers of the KIR2DS2 gene, the HLA-B*41 allele, the KIR2DS1 + HLA-C2 pairincreased risk in non-carriers of KIR2DL3 + HLA-C1/C2 pair or the KIR2DL1 + HLA-C1/C2 pair [128]	aOR = 27.22, P = 0.032; aOR = 68.84, P = 0.033; aOR = 28.58, P = 0.024 aOR = 43.04, P = 0.034; aOR = 43.04, P = 0.034
CNS TB-IRIS	Cerebrospinal fluid (CSF) neutrophil count	CSF neutrophilic predominance predicted IRIS	AUC 0.72, 95% CI 0.54–.090 [129]
	CSF culture positive	TBM patients with positive cultures had higher IRIS rates	RR 9.3, 95%CI 1.4–62.2 [129]
	Elevated CSF TNF-α, low CSF IFN-λ	Together predict IRIS development in TBM	AUC 0.91, (CI 0.53–0.99) [129]
Un-Masking IRIS	Lymphadenopathy on chest radiograph	Study does include other unmasked opportunistic infections [41]	aHR 9.15;(CI 4.10–20.42)
	Anemia	Hemoglobin <10 vs. >12 g/dL [41]	aHR 3.36, (CI 1.32–8.52)
	Elevated CRP	CRP ≥ 25 vs. <25 mg/L [41]	aHR 2.77, (CI 1.31–5.85)
	Weight loss	≥10% vs. <10% weight loss prior to ART [41]	aHR 2.31, (CI 1.05–5.11)

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
