# Peer review of "Tuberculosis IRIS: Pathogenesis, Presentation, and Management across the Spectrum of Disease"

_life, 2020, doi:10.3390/life10110262_

Round 1

Reviewer 1 Report

This manuscript thoroughly described the TB-associated immune reconstitution inflammatory syndrome (TB-IRIS) during TB and HIV co-infection. It is very well written in English and includes large amount of detailed information about TB-IRIS. I will recommend acceptance after minor revision. The only comment I have is the figure order. Figure 1 and 2 should be switched. Thanks!

Reviewer 2 Report

They report on TB-IRIS.
The content of the article is a detailed review of the current pathogenesis of IRIS, and it seems to be OK.

The only thing I would point out is that the position of Figure 1 is later than that of Figure 2.

Thank you for this valuable review.
I look forward to seeing it published.

Reviewer 3 Report

  1. “……cold abscesses, worsening radiological features, worsening serositis, 61 or worsening of symptoms of at least two diseases locations (i.e. constitutional symptoms, respiratory 62 symptoms, abdominal symptoms)” Please include references.
  2. “Examples include but are not limited to lymphadenitis, abscesses, 68 respiratory failure, or a systemic inflammatory syndrome. Importantly, in this case TB was not 69 diagnosed or treated prior to initiation of ART” Reference is missing.
  3. “commonly reported signs albeit with wide reported ranges, present in 5.6 to 88.5%, and 30% - 100% 94 of cases respectively” The authors cite review by Namale et al. Ideally, they should cite the original article rather than review article to be more accurate.

  1. “While TB-IRIS is known to be associated with profound elevations in a number of cytokines and 96 inflammatory markers, only C-reactive protein (CRP) is readily available for clinical use” Sentence is not clear.

  1. “Evidence of immune reconstitution further supports the diagnosis of IRIS, and a repeat CD4 count is important for diagnosis.” What is repeat CD4 count ?

  1. “Endemic non-opportunistic infections such as malaria often cause fever and should be considered based on the patient’s location” Malaria related Fever is clinically distinct and TB produce variety of symptoms other than Fever so I am not sure what is the rational here.

  1. “The prognosis of TB-IRIS varies based on site of disease and treatment protocols, but it is most 134 frequently self-limited” Sentence is not clear.

  1. “Approximately half of HIV patients with pulmonary TB initiated on ART experienced clinically significant declines in FEV1 function, and 18% experienced severe declines at six months treatment” Please improve sentence.

  1. “TB-IRIS results from a disproportionate and dysregulated inflammatory response to TB produced as a consequence of rapid recovery of the immune system. The immunology of HIV-TB coinfection has been reviewed extensively” Authors should provide more background on TB pathogenesis and different model systems that can be used for IRIS (PMID: PMID: 28137237, 30082569,  31752895)

  1. “In addition, decreased TNF- release compromises macrophage apoptosis (an important host-defense mechanism against TB)[52] and Toll-like receptor (TLR) signaling involved in TB recognition is decreased as well [53].” Please explain the mechanism by which TNF-a compromise macrophage apoptosis.
  2. “For example, CD4 T cell count may increase, but subset ratios remain unbalanced [55].” Which subset ratio ?
  3. “Activation of pro-inflammatory elements of the complement system also appear involved in the exuberant macrophage response, based increased expression of related genes, including C5” Sentence not clear.
  4. “In resource limited settings, urine tests for the TB cell wall lipoprotein TB-lipoarabinomannan (LAM) may assist in risk-stratification, as LAM positivity is associated with both low CD4 count and disseminated TB [110].” What is LAM positivity. Authors have mentioned that humoral response don’t play a role in IRIS ?

Round 2

Reviewer 3 Report

The authors have answered all of my queries and made changes to the review manuscript. I recommend acceptance of this review article.